# Mutual Influence of Human Cytochrome P450 Enzymes and UDP-Glucuronosyltransferases on Their Respective Activities in Recombinant Fission Yeast

**DOI:** 10.3390/biomedicines11020281

**Published:** 2023-01-19

**Authors:** Sangeeta Shrestha Sharma, Shishir Sharma, Jie Zhao, Matthias Bureik

**Affiliations:** School of Pharmaceutical Science and Technology, Tianjin University, 92 Weijin Road, Nankai District, Tianjin 300072, China

**Keywords:** Cytochrome P450, UDP-glucuronosyltransferase, CYP2C9, CYP2D6, CYP4Z1, UGT2B7

## Abstract

Cytochromes P450 (CYPs) and UDP-glucuronosyltransferases (UGTs) are the most important human drug metabolizing enzymes, but their mutual interactions are poorly understood. In this study, we recombinantly co-expressed of each one of the 19 human members of the UGT families 1 and 2 with either CYP2C9, CYP2D6, or CYP4Z1 in fission yeast. Using these strains, we monitored a total of 72 interactions: 57 cases where we tested the influence of UGT co-expression on CYP activity and 15 cases of the opposite approach. In the majority of cases (88%), UGT co-expression had a statistically significant (*p* < 0.05) effect on P450 activity (58% positive and 30% negative). Strong changes were observed in nine cases, including one case with an activity increase by a factor of 23 (CYP2C9 activity in the presence of UGT2A3) but also four cases with a complete loss of activity. When monitoring the effect of CYP co-expression on the activity of five UGTs, activity changes were generally not so pronounced and, if observed, always detrimental. UGT2B7 activity was not influenced by CYP co-expression, while the other UGTs were affected to varying degrees. These data suggest the notion that mutual influence of CYPs and UGTs on each other’s activity is a widespread phenomenon.

## 1. Introduction

A major biotechnological application of human enzymes is the biocatalysis of chemical reactions that take place during the metabolism of drugs and xenobiotics [1]. The metabolism of foreign compounds in humans can be subdivided into two phases. In Phase I (“functionalization phase”), a new functional entity such as a hydroxy group is introduced into a xenobiotic compound, while in Phase II (“conjugation phase”), xenobiotics or Phase I metabolites are conjugated with a charged species such as glucuronic acid or sulfate. The most important Phase I enzyme family is that of the cytochromes P450 (CYPs or P450s), while UDP-glucuronosyltransferases (UGTs) constitute the most important Phase II enzymes [2]. CYPs are a large superfamily of monooxygenases that are present in all biological kingdoms [3] and were named after the characteristic peak at 450 nm observed in reduced CYPs in complex with carbon monoxide [4]. Typically, P450s cannot be reduced by nicotinamide adenine dinucleotide phosphate (NADPH) themselves but are the terminal oxidase enzymes in short electron-transfer chains, which include one or two electron transfer proteins; in humans, these are the microsomal cytochrome P450 reductase (CPR or POR) or the mitochondrial proteins adrenodoxin reductase (AdR) and adrenodoxin (Adx), respectively [5]. The 57 members of the human CYP family are membrane-bound enzymes that are typically located either on the cytoplasmic side of the endoplasmic reticulum or on the matrix side of the inner mitochondrial membrane, but they are sometimes also found in the nuclear envelope and on the plasma membrane [6,7,8]. Human CYPs metabolize a huge variety of substrates of many different chemical classes, and while they are best known for catalyzing hydroxylation reactions, some can also perform dealkylations at heteroatoms, sulfoxidations, epoxidations, deaminations, and even other reaction types [1,9]. UGTs catalyze the conjugation of glucuronic acid (or other sugars) to substrates with polar groups (such as hydroxyls, thiols, carboxyls, or amines); and like CYPs, the 22 human UGT enzymes can metabolize a wide variety of lipophilic endogenous molecules, environmental compounds, and synthetic drugs [10,11]. These human UGTs reside in the luminal side of ER [12] and are classified into four families: UGT1, UGT2, UGT3, and UGT8. The UGT1 family has nine members (UGT1A1, 1A3, 1A4, 1A5, 1A6, A7, 1A8, 1A9, and 1A10), while the UGT2 family consists of the two sub-families UGT2A (UGT2A1, 2A2, and 2A3) and UGT2B (UGT2B4, 2B7, 2B10, 2B11, 2B15, 2B17, and 2B28); in addition, the UGT3 family has two members (UGT3A1 and 3A2), and UGT8A1 is the sole member of the UGT8 family. The main cofactor of UGTs is uridine diphosphate glucuronic acid (UDPGA), which is formed by UDP glucose-6-dehydrogenase (UGDH) through oxidation of UDP glucose [13]. In this connection, it is important to remember that the functions of both enzyme families are not limited to drug metabolism alone, as they play important roles in many other physiological processes that are independent of xenobiotics [10,14].

A number of CYPs and UGTs are predominantly expressed in the liver, which of course is the most important organ for drug metabolism. However, to date, the interactions between the members of the two families are poorly understood. UGTs form both homo- and hetero-oligomers, and such UGT–UGT interactions affect their enzymatic activities [15]. Mutual association between some CYPs and UGTs was repeatedly reported in coprecipitation experiments using liver samples from humans [16,17,18] and rats [19]. However, an untargeted mass spectrometry analysis of affinity-purified UGT1A enzymes and associated protein complexes in liver, kidney, and intestine tissues did not point towards CYPs being among the most important UGT interaction partners [20]. However, it cannot be excluded that such interactions are at least in some cases of a rather transient nature, which would impede their experimental verification. In summary, the endoplasmic reticulum of hepatic cells (and to a lesser extent also cells of other tissues) contains multiple CYP and UGT proteins, and it is reasonable to assume that, in principle, they might interact. From a biotechnological point of view, such interactions are, in principle, either beneficial, harmful, or neutral with respect to the space-time yield.

Reports on the consequences of CYP-UGT interactions on the activities of the enzymes involved show that such effects depend (among other factors) on the specific enzymes involved [15,21]. It has been suggested that some CYPs and UGTs form a metabolosome (a functional unit of metabolism consisting of multiple metabolism-related proteins) [18]. Moreover, in addition to the common case where CYP metabolites are UGT substrates, this network of drug metabolism becomes even more complex by the fact that Phase I and Phase II reactions do not necessarily occur in sequential order, as many drugs (for instance, olanzapine [22]) can directly be metabolized by both CYPs and UGTs. Hence, it is far from easy to specifically determine the effect of CYPs on UGT activity and vice versa.

Metabolites of xenobiotic compounds are important in many areas, for instance, in the assessment of drug-caused side effects or toxicity and in doping analysis. In the synthesis of such metabolites, chemical approaches and biological strategies compete. Establishing functional, versatile, and efficient cascades of drug-metabolizing enzymes (DMEs) should constitute a major step towards the overall goal of making biocatalysis more attractive for such applications. However, a necessary prerequisite for the construction of such cascades is an understanding of the influence that the co-expressed enzymes might display on each other’s activities. Therefore, it was the aim of the present study to determine the effect of co-expression of each one of the 19 human members of the UGT families 1 and 2 on the activity of three human CYPs. In this way, two of the most important drug metabolizing CYPs (CYP2C9 and CYP2D6) could be tested with CYP4Z1 as an internal control (as it is not known to have a function in xenobiotic biotransformation, and many of the marketed and in-house synthesized luciferin probe substrates are metabolized by this enzyme [23,24,25,26], which makes it a reliable control for bioluminescent assay in CYP metabolism).

## 2. Materials and Methods

### 2.1. Chemicals and Reagents

Chemicals and reagents were obtained from the same sources as before [27]. Glucose, disodium hydrogen phosphate, potassium dihydrogen phosphate, ammonium chloride, and potassium hydrogen phthalate were purchased from Chemart Chemical (Tianjin, China). Biotin, agar, calcium chloride monohydrate, citric acid, copper sulphate, ferric chloride, boric acid, potassium chloride, potassium iodide, manganese sulphate, magnesium chloride, molybdenum oxide, adenine, histidine, leucine, uracil, sodium pantothenate, nicotinic acid, thiamine, and zinc sulfate heptahydrate were from Kermel Chemical (Tianjin, China); Tris-HCl was from AKZ-Biotech (Tianjin, China), and yeast extract, yeast nitrogen base, malt extract, and glycerol were from Dingguo (Tianjin, China); Triton-X100 was from Leagene (Beijing, China); white 96-well microtiter plates were from Nunc (Thermo-fisher scientific, Lagenselbold, Germany). The NADPH regeneration system, Luciferin-H (probe for CYP2C9), Luciferin-ME EGE (probe for CYP2D6), Light Detection reagent (LDR), Light Detection reagent with Esterase (LDRE), UGT-Glo substrate A, UDP-glucuronic acid, and d-cysteine were all from Promega (Madison, USA). Luciferin-4FBE (synthesized in-house as described [23]) was used as a probe substrate for CYP4Z1. All other chemicals and reagents used were of the highest grade available.

### 2.2. Fission Yeast Strains and Media

Preparation of media and general techniques were completed as described [28]. Briefly, strains were generally cultivated at 30 °C in Edinburgh Minimal Medium (EMM) with supplements of 0.1 g/L final concentration as required. Liquid cultures were maintained shaking at 150 rpm. Thiamine was used at a concentration of 5 μM. Parental strains JMN11 and JMN12 were obtained by mating of strain MB175 (genotype h- ade6.M210 leu1-32 ura4-D18 his3.Δ1) [29] with strain MW5 (genotype h+ ade6.M216 ura4-D18; kind gift of Rudolf Drescher) and subsequent selection of clones with the desired markers and mating types [30]. All strains used in this study are listed in Appendix A.

### 2.3. Preparation of Enzyme Bags

Enzyme bags were prepared as described [27]. Briefly, strains were streaked on EMM plates with required supplements and incubated at 30 °C for 3 days, then cultured in 10 mL EMM broth at 230 rpm and 30 °C for 36 h. After counting the cell in a Neubauer improved hemocytometer counting chamber, 5 × 10^7^ cells per reaction were pelleted (5000× *g*, 4 °C for 5 min) and then permeabilized by 0.3% Triton-X 100 in Tris-KCl buffer (100 mM Tris-base, 200 mM KCl, pH 7.8) at 30 °C for 1 h at 230 rpm. After three washings with chilled 50 mM NH_4_HCO_3_, 100 µL phosphate-buffered saline (PBS) (8 g NaCl, 0.2 g KCl, 1.44 g Na_2_HPO_4_, 0.24 g KH_2_PO_4_, 0.133 g CaCl_2_ • 2 H_2_O, 0.1 g MgCl_2_ • 2 H_2_O in 100 mL H_2_O, pH 7.4) with 50% glycerol was added to 5 × 10^7^ cells per tube, mixed with gentle vortexing, and the mixture aliquoted in 1.5 mL Eppendorf tubes. The tubes were flash-frozen in liquid nitrogen and stored at −80 °C until further use.

### 2.4. Bioluminescence Assay for CYP Activity of Diploids

Next, 100 µL of the permeabilized cells (enzyme bags) stored at −80 °C were thawed in ice, washed with an equal volume of 100 mM potassium phosphate buffer (pH 7.4), and then processed further for activity assays as described previously [27]. In brief, the proluciferin substrates Luciferin-H (100 µM), Luciferin-MEEGE (10 µM), and Luciferin-4FBE (150 µM) were used for CYP2C9, CYP2D6, and CYP4Z1, respectively. CYP activity assays were performed in 100 mM potassium phosphate buffer (pH 7.4) at 37 °C and 1000 rpm for 3 h. The supernatant after centrifugation was transferred to a white, opaque, 96-well plate, and an equal volume of LDR or LDRE was added depending on the substrate used. After the reaction was stabilized for 20 min at room temperature, the reading was taken as relative light units (RLU) in an INFINITE M200 Pro luminometer from TECAN. The readings were then converted to picoMoles (pM) per minutes using a d-luciferin calibration curve. All reactions were performed three times in triplicate.

### 2.5. Bioluminescence Assay for UGT Activity of Diploids

UGT activity assays were performed in a similar way as CYP activity assays described above except for the probe substrate (UGT-Glo assay kit from Promega). Activity assays were performed as described previously [27] with 20 µM UGT multi-enzyme substrate (UGT-Glo substrate A) and 4 mM UDPGA in 40 µL reaction volume in 50 mM TES buffer.

### 2.6. Statistical Analysis

All data were calculated from experiments performed at least thrice in triplicate and are presented as mean ± standard deviation. Statistical significance was determined using a two-tailed *t*-test. Differences were considered significant if *p* < 0.05. Statistical analysis was completed using GraphPad Prism 5.01 (GraphPad Software Inc., La Jolla, CA, USA).

## 3. Results and Discussion

### 3.1. Construction of Yeast Strains

Human DMEs are attractive biocatalysts [1], and the fission yeast *Schizosaccharomyces pombe* was previously shown to be a good model system for the recombinant expression of human CYPs, SULTs, and UGTs [31,32,33]. However, so far, it was never attempted to co-express both Phase I and Phase II enzymes within the same strains. Such a co-expression of different DMEs within the same microbe offers new and interesting perspectives for many applications, such as exploring new chemical space by library screening or efficient synthesis of drug metabolites, which require multiple reaction steps by cascade design [34]. However, the mutual effects of co-expression on the performance of the individual enzymes need to be considered. In this study, we therefore co-expressed human CYPs and UGTs in fission yeast in order to explore such effects.

In previous investigations, we successfully employed the expression vectors pCAD1 and pREP1 for the recombinant expression of UGTs and CYP systems in fission yeast [31,35]. pCAD1 allows for targeted integration into the leu1 gene of *S. pombe* [36], while pREP1 is an autosomal replicating plasmid that bears the LEU2 gene from *S. cerevisiae*, which can complement leu1 defects in fission yeast [37]. Both expression vectors harbor the strong endogenous nmt1 promotor, which permits regulation of foreign gene expression via the presence or absence of thiamine in the culture medium [38]. As each of these vectors can only be used once in any given haploid fission yeast strain, we chose to create diploid strains for the co-expression of three human genes (CPR with a CYP and a UGT). In these diploid strains, one leu1 allele is disrupted by pCAD1-CPR and the other one by pCAD1-UGT; moreover, the complementing markers ade6-M210 and ade6-M216 were used [39]. An overview of the cloning strategy is given in Figure 1, and all strains are listed in Appendix A. The haploid parental strain JMN12 (genotype h- ade6-M216 ura4-D18 his3.Δ1) was transformed either with the integrative vector pCAD1 or with one of 19 derivatives of pCAD1 that contain one of the human UGT genes to yield 20 new strains (SAN3 to SAN22). Another haploid parental strain, JMN11 (h+ ade6-M210 ura4-D18 his3.Δ1) [29], was transformed either with the integrative vector pCAD1 or with pCAD1-CPR to yield the new strains SAN1 (h+ ade6-M210 ura4-D18 his3.Δ1 leu1::pCAD1) and SAN2 (h+ ade6-M210 ura4-D18 his3.Δ1 leu1::pCAD1-CPR). Mating of SAN2 with each one of strains SAN3 to SAN22 yielded a set of 19 diploid strains that co-express CPR with one of the UGTs and a diploid control strain that only expresses CPR (SAN100 to SAN119). Finally, transformation of the diploid strains SAN100 to SAN119 with an expression plasmid for a CYP enzyme (e.g., pREP1-CYP2C9) yielded three new series of diploid strains that co-express CPR together with a CYP and a UGT (including controls); these strains are SAN200 to SAN219 (CYP2C9 series), SAN300 to SAN319 (CYP2D6 series), and SAN500 to SAN519 (CYP4Z1 series), respectively. In addition, mating of SAN1 with SAN3 yielded the new strain SAN120 (diploid control for UGT activity), and mating of SAN1 with SAN6, SAN11, SAN15, SAN17, or SAN22, respectively, yielded the new UGT diploid strains SAN123 (UGT1A4), SAN128 (UGT1A9), SAN132 (UGT2A3), SAN134 (UGT2B7), and SAN139 (UGT2B28), respectively, which were used for UGT activity assays. Thus, a total of 24 new haploid and 86 new diploid fission yeast strains were cloned in this study.

### 3.2. Influcence of UGT-Coexpression on P450 Activity

In order to monitor the possible effect of UGTs on the activity of CYP2C9, strains SAN200 to SAN219 were used for the preparation of enzyme bags. Activity of these enzyme bags towards Luciferin-H was then monitored as described previously [25]. Interestingly, co-expression of 16 UGTs lead to statistically significant increases in CYP2C9 activity (*p* < 0.005), while two lead to an activity decrease (*p* < 0.05), and one (UGT1A6) had no effect (Figure 2). Activity increases varied between 1.7 fold (UGT1A3 and UGT1A5) and 23 fold (UGT2A3), with UGT1A10 causing the second highest increase (11 folds). Reduction of activity was not very pronounced with UGT1A7 (factor 1.6), whereas the most striking effect was seen upon co-expression of UGT2B7, which led to a complete loss of CYP2C9 activity (*p* < 0.001).

Diploid fission yeast strains co-expressing CYP2D6 and UGTs were tested for CYP2D6 activity in the same way as described for CYP2C9 (see above) except that the substrate Luciferin-ME EGE was used. In this case, co-expression of six UGTs led to a statistically significant increase in P450 activity (*p* < 0.05), while eleven caused an activity decrease (*p* < 0.05), and the remaining two had no effect (Figure 3). Activity increases varied between 1.4 fold (UGT2B10) and 4.3 fold (UGT1A4), while reduction of activity was also moderate in nine out of the eleven cases (factor 1.3 to 3.6). Once more, the strongest effect was seen upon co-expression of UGT2B7, which again caused a complete loss of activity, while UGT1A8 reduced CYP2D6 activity by a factor of 33.

Strains co-expressing CYP4Z1 and UGTs were tested for CYP activity using the substrate Luciferin-4FBE. In these experiments, co-expression of eleven UGTs led to statistically significant (*p* < 0.05) activity increases, while four caused an activity decrease (*p* < 0.005), and another four had no effect (Figure 4). Interestingly, activity increase factors were higher than seen with CYP2D6, spanning a range of 1.3 (UGT1A5) to 15 (UGT2B28); a strong increase was also seen for UGT2B10 (14 folds). With respect to activity reduction, co-expression of two UGTs (1A6 and 1A7) had a moderate effect (factors 2.4 and 2.7), while UGT1A3 reduced CYP4Z1 activity by a factor of eleven, and UGT1A10 completely abolished this activity. In view of the findings reported above, it was unexpected that co-expression of UGT2B7 did not have any effect on CYP4Z1 activity.

### 3.3. Influcence of P450-Coexpression on UGT Activity

Next, we decided to test for a possible effect of CYP co-expression on the activity of five UGTs (UGT1A4, 1A9, 2A3, 2B7, and 2B28). Again, a luciferin-based probe substrate was used (UGT-Glo substrate A). It was found that the activities of UGT1A4, UGT1A9, and UGT2B28, respectively, were always reduced (*p* < 0.05) upon co-expression with each of the three CYPs tested (Figure 5), while UGT2B7 activity was not affected at all. In the case of UGT2A3, only CYP4Z1 co-expression led to reduced activity, while the other two CYPs had no statistically significant influence (*p* > 0.05). Activation of UGT activity was not observed for any UGT-CYP combination. In conclusion, these data show that co-expression of CYPs leads to differential effects on UGT activity, just as UGT co-expression affects CYP activities (see above).

With respect to the tendency of the previously reported interactions, we did not find a negative effect of CYP2C9 co-expression on UGT2B7 activity as was reported previously [40]. However, in view of the very different experimental approaches (expression systems, substrates), we consider this outcome to be within reasonable expectations; for comparison, varying effects of UGT2B7 on CYP3A4 activity were observed by the same group depending on whether microsomes or liver homogenates were used [41,42].

Of course, it would be interesting to determine the mutual effects of all human CYPs and UGTs on each other’s activities. However, the large numbers involved prevented us from cloning more than 1000 diploid strains and performing several 10,000 individual activity assays. Prior to this study, only eight such interactions had been reported in the literature: Co-expression of either UGT1A9 or UGT2B7 was shown to impede CYP3A4 activity, while CYP3A4 itself changed the activity of UGT1A1, UGT1A6, UGT1A7, and UGT2B7, respectively; moreover, UGT2B7 activity was also reported to be hampered in the presence of CYP1A2 and CYP2C9 [40,41,42,43,44,45]. In the present work, we present data for a total of 72 interactions: 57 cases of the influence of UGT co-expression on CYP activity and 15 cases of the opposite approach. Notably, most previous studies on the mutual influence of CYPs and UGTs on each other’s activity reported either no effect or a negative effect. Here, we present evidence for 33 positive, 12 neutral, and 27 negative DME–DME interactions. Remarkably, a positive effect of co-expression was only observed for CYP activities but not for those of UGT; however, this might change when more UGTs are tested in this type of assay.

The data in our study, as those of most previous works in this field of research, were generated using the most standard variations of the enzymes involved (also called .1-variations). However, it is reasonable to assume that at least in some cases of CYP-UGT interactions, polymorphic variants of either enzyme might have an influence on such interactions [42]. It is also conceivable that in some instances, polymorphic variants might show interactions, while the .1-variants do not. However, that might not be true for all variants of these two enzymes. Interestingly, hetero-oligomers of two murine UGTs have recently been shown to catalyze glucuronidation reactions that are not observed with homo-oligomers of either UGT [44], thus pointing towards still another way of protein–protein interactions influencing DME activities.

As a summary, it can be said that co-expression of DMEs within the same host is promising in a number of cases but not indicated in others. In the latter cases, coculture of two strains that separately express the respective enzymes might be a more promising biocatalytical approach. There are still many facets of CYP-UGT interactions that need to be explored. It seems to the authors of this work that currently, we are just looking at the tip of the iceberg.

## 4. Conclusions

In this study, we provide by far the most comprehensive investigation into interactions between human CYPs and UGTs upon co-expression within the same microbe. In 50 out of 57 cases tested (= 88%), UGT co-expression had a statistically significant effect (*p* < 0.05) on P450 activity; in 33 instances, the effect was positive, while in the other 17 cases, it was detrimental. Strong changes of activity (i.e., by more than a factor of ten) were observed in nine cases: four positive and five negatives, including three with a complete loss of activity. Altogether, we present evidence for 33 positives, 12 neutral, and 27 negative interactions involving CYP and UGT activity. Positive effects were only observed for CYP activities upon UGT co-expression but never the other way around. In summary, our data highlight some of the opportunities and pitfalls that may result from the co-expression of human CYPs and UGTs within the same microbial host.

## Figures and Tables

**Figure 1 biomedicines-11-00281-f001:**
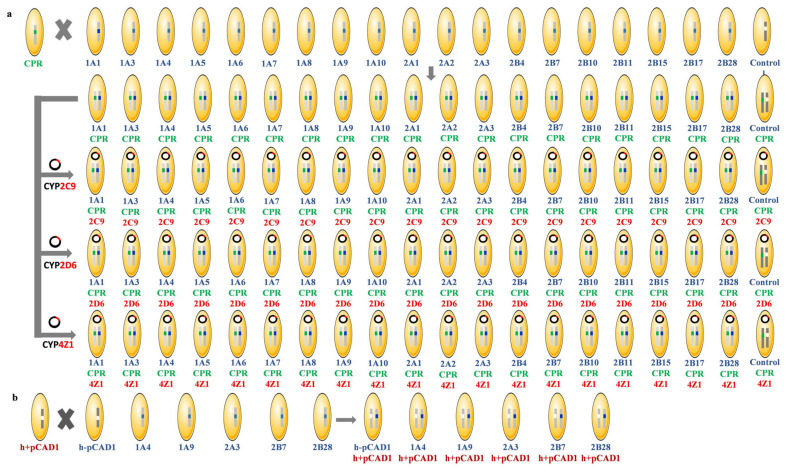
Cloning Strategy. (**a**) Parental haploid strain JMN12 (mating type h-) was transformed with the integrative vector pCAD1 and with 19 derivatives of pCAD1 that contain one of the human UGT genes to yield 20 new strains (SAN3 to SAN22; top, marked blue). Parental haploid strain JMN11 (mating type h+) was transformed with pCAD1-CPR to yield the new strain SAN2 (top, marked green). Mating of these strains yielded 19 new diploid strains that co-express CPR, with one of the UGTs plus a control that only expresses CPR (SAN100 to SAN119; top right). These 20 new diploid strains were then individually transformed with three different autosomal replicating plasmids (pREP1-CYP2C9, pREP1-CYP2D6, and pREP1-CYP4Z1, respectively; marked red) to yield three new series of diploid strains that co-express CPR together with a CYP and a UGT (including controls); these strains are SAN200 to SAN219 (CYP2C9 series), SAN300 to SAN319 (CYP2D6 series), and SAN500 to SAN519 (CYP4Z1 series), respectively. (**b**) Parental haploid strain JMN11 was transformed with pCAD1 yielding SAN1. Mating of this strain with SAN3, SAN6, SAN11, SAN15, SAN17, and SAN22 yielded the new strains SAN120 (no insert), SAN123 (UGT1A4), SAN128 (UGT1A9), SAN132 (UGT2A3), SAN134 (UGT2B7), and SAN139 (UGT2B28) used as diploid controls for UGT activity assays.

**Figure 2 biomedicines-11-00281-f002:**
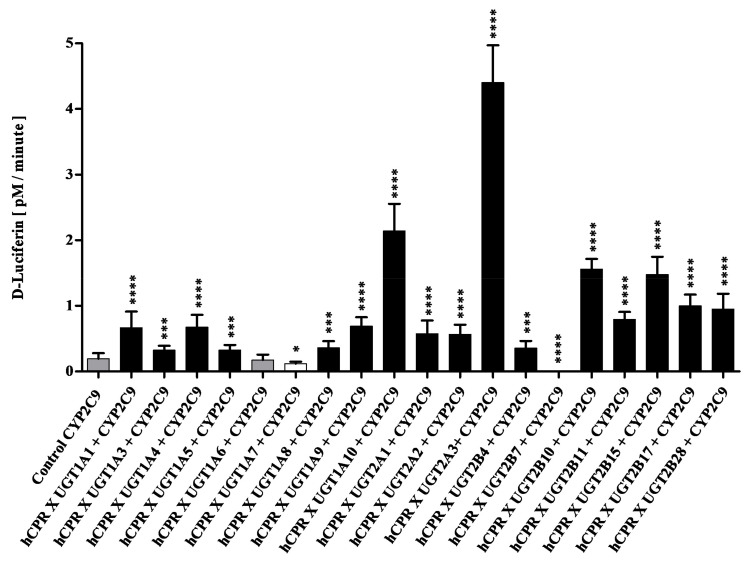
Enzymatic activities of enzyme bags prepared from diploid strains co-expressing CYP2C9, CPR, and different UGTs as indicated. Activities towards the substrate Luciferin-H are shown in comparison to the control strain SAN200 (which co-expresses CYP2C9 and CPR without a UGT). Activities that are not different from the control are shown in grey, while those that are statistically significantly higher or lower than the control are shown in black or white, respectively. * *p* < 0.05, *** *p* < 0.005, **** *p* < 0.001.

**Figure 3 biomedicines-11-00281-f003:**
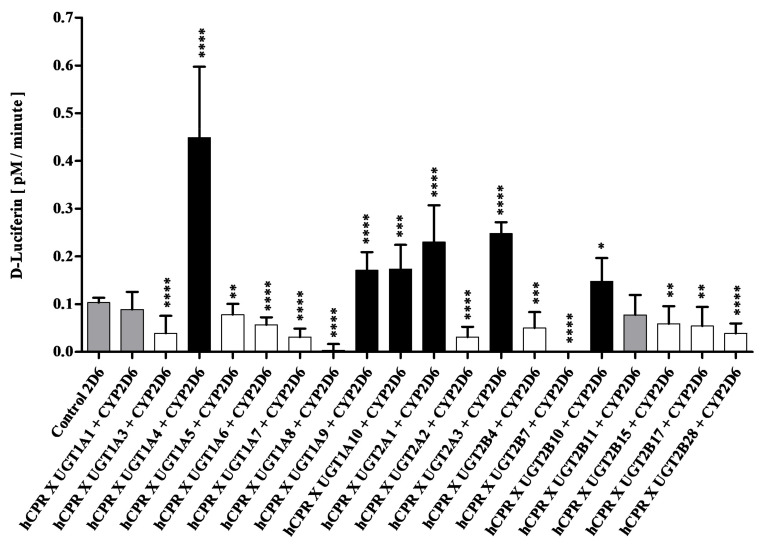
Enzymatic activities of enzyme bags prepared from diploid strains co-expressing CYP2D6, CPR, and different UGTs as indicated. Activities towards the substrate Luciferin-MEEGE are shown in comparison to the control strain SAN300 (which co-expresses CYP2D6 and CPR without a UGT). Activities that are not different from the control are shown in grey, while those that are statistically significantly higher or lower than the control are shown in black or white, respectively. * *p* < 0.05, ** *p* < 0.01, *** *p* < 0.005, **** *p* < 0.001.

**Figure 4 biomedicines-11-00281-f004:**
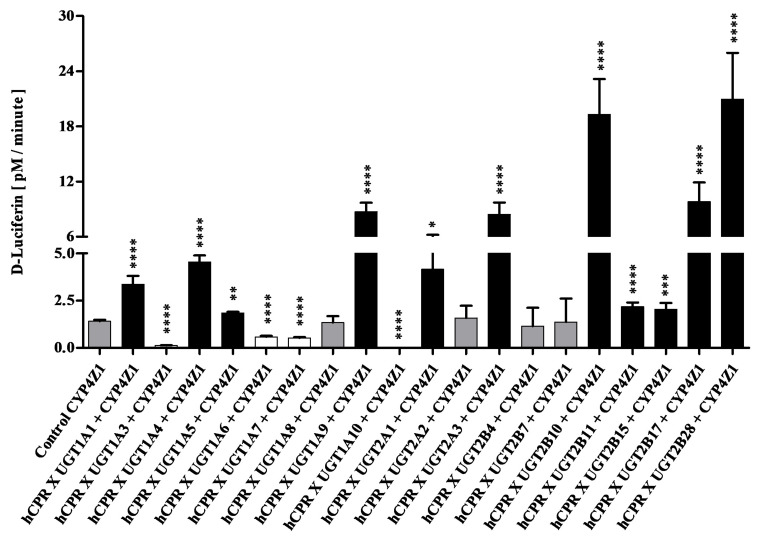
Enzymatic activities of enzyme bags prepared from diploid strains co-expressing CYP4Z1, CPR, and different UGTs as indicated. Activities towards the substrate Luciferin-4FBE are shown in comparison to the control strain SAN500 (which co-expresses CYP4Z1 and CPR without a UGT). Activities that are not different from the control are shown in grey, while those that are statistically significantly higher or lower than the control are shown in black or white, respectively. * *p* < 0.05, ** *p* < 0.01, *** *p* < 0.005, **** *p* < 0.001.

**Figure 5 biomedicines-11-00281-f005:**
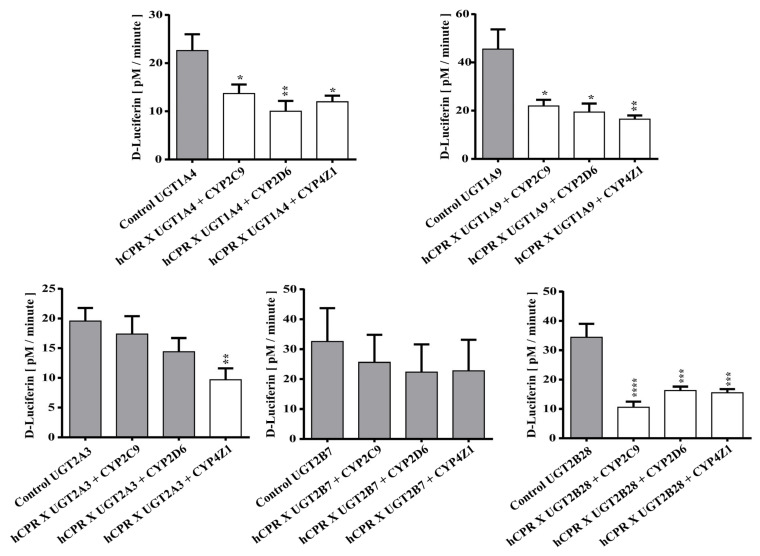
Enzymatic activities of enzyme bags prepared from diploid strains co-expressing five different UGTs with CPR and four different CYPs as indicated. Activities towards UGT-Glo substrate A are shown in comparison to the respective control strains (which only express one of the UGTs). Activities that are not different from the control are shown in grey, while those that are statistically significantly lower than the control are shown in white. * *p* < 0.05, ** *p* < 0.01, *** *p* < 0.005, **** *p* < 0.001.

## Data Availability

The datasets generated and/or analyzed during the current study are available from the corresponding author on reasonable request.

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
