# Peer review of "Mutual Influence of Human Cytochrome P450 Enzymes and UDP-Glucuronosyltransferases on Their Respective Activities in Recombinant Fission Yeast"

_biomedicines, 2023, doi:10.3390/biomedicines11020281_

Round 1

Reviewer 1 Report

The main concept of the paper is really interesting and worth noticing. However, some improvements are strongly recommended, i.e.:

1. Section 2.1.: this section should provide adequate information concerning the origin of the reagents applied as well as brief description of the synthesis methodology of Luciferin-4FBE.

2. Section 2.2.: both the methodology of the enzyme bags preparation and the activity assays need to be briefly described.

3. Final conclusions should be supplemented with some quantified data.

4. Section References should contain significantly more up-to-date publications.

Reviewer 2 Report

In this submitted manuscript, Dr. Bureik and co-authors systematically evaluated the mutual interactions of P450 enzymes and UDP-glucuronosyltransferases (UTG) when they coexpressed in fission yeast. Using the strains that the authors designed, a total of 72 interactions were observed, where 57 cases tested the influence of UGT coexpression on CYP activity and 15 cases of the opposite approach. It was found that UTG coexpression had a statistically significant effect on P450 activity. While in the opposite approach, activity changes were generally not so pronounced, and if any, are always detrimental.

This manuscript is well-written and of high quality. It not only shows the sufficient background and the area that hasn’t been explored in this study, but also provides reasonable project design and analyzes the data professionally. Interesting results were found, and they pave the way for future investigations. The science reported in this manuscript is sound, but the format of the writing would need some revisions. In terms of the manuscript content and importance, the work would appeal to the broad readership of Biomedicines and I would recommend accepting it for publication after minor revisions.

  1. It is recommended to merge sections 3 and 4 into one as Results and Discussion. In this case, the discussion about the experimental data can be followed in each subcategory, and make the manuscript more compact and easy to follow.
  2. On line 5, Dr. Bureik should have the affiliation with institute/address 1, instead of 2. Also, if there is institute/address 2, it should be provided.
  3. On line 7, the [email protected] should be deleted. It makes no sense given here.
  4. Line 37, the full name of NADPH should be given when it appears for the first time.
  5. Line 127, it should be “Supporting Information” instead of “Supporting Informations”. The same revision should be done with the supporting information document.
  6. Lines 165, 166, 180, 181, 197, and 245, the word “fold” should be “folds”, and a space should be left between the number and “folds”, like “1.7 folds”.
  7. Line 170, no need to bolden the word “CYP2C9”.
  8. Line 194, the sentence should be “…statistically significant activity increases…”, rather than “…statistically significantly activity increases…”. The same revision should be done for the sentence on lines 247 and 251.
  9. One extra dot was found on line 292 as “…while the .1-variants do not”.
  10. Line 306, the word “twelve” is better to be “12” to keep it consistent with others. 

Reviewer 3 Report

The manuscript needs majopr edits in the methodology section. The authors used ref 22 for the manuscript but changes in the results in synthesis, activity assays and control starin comparison activity could be different. Please include the methods in brief will be useful for the readers.

In line #14, UGT coexpression 14 had a statistically significant effect on P450 activity (58% positive and 30% negative). Write the level of significance for the comparison.

Line#20,  yarying degrees. Please check for typo error.

Write a brief note on UGT families in the introduction.

Is there any specific reason for selection of CYP2C9 and CYP2D6, CYP4Z1 as an internal control in the study?

In material section, write the Luciferin-4FBE synthesis yield/

Most of the studies are linked to ref 22. Please briefly elaborate the methods and activity assay results of enzyme bags in the manuscript.

Write the serial number column in the Table S1. Is there any possibility to differentiate or separate the column as per parental strain (None, JMN, SAN).

Round 2

Reviewer 1 Report

The paper has been significantly improved thus in my opininon it may be accepted for publication in the Journal.

Reviewer 3 Report

No further comments.